# Comprehensive Transcriptome Analysis of Follicles from Two Stages of the Estrus Cycle of Two Breeds Reveals the Roles of Long Intergenic Non-Coding RNAs in Gilts

**DOI:** 10.3390/biology11050716

**Published:** 2022-05-06

**Authors:** Mingzheng Liu, Qinglei Xu, Jing Zhao, Yanli Guo, Chunlei Zhang, Xiaohuan Chao, Meng Cheng, Allan P. Schinckel, Bo Zhou

**Affiliations:** 1College of Animal Science and Technology, Nanjing Agricultural University, Nanjing 210095, China; liumingzheng@stu.njau.edu.cn (M.L.); 2019205004@njau.edu.cn (Q.X.); 2019105039@stu.njau.edu.cn (J.Z.); 2019105040@njau.edu.cn (Y.G.); 2020105039@stu.njau.edu.cn (C.Z.); 2021205020@stu.njau.edu.cn (X.C.); 2020805124@stu.njau.edu.cn (M.C.); 2Department of Animal Sciences, Purdue University, West Lafayette, IN 47907-2054, USA; aschinck@purdue.edu

**Keywords:** estrus expression, lincRNAs, follicles, pigs, RNA-seq

## Abstract

**Simple Summary:**

This study provides new perspectives about the roles of lincRNAs in the estrus expression of gilts, which is correlated with ovarian steroid hormone and follicular development. Follicular tissues from two stages of the estrus cycle of Large White and Mi gilts were used for RNA-seq. Some genes and lincRNAs related to estrus expression in pigs were discovered. PPI and ceRNA networks related to the estrus expression were constructed. These results suggest that the estrus expression may be affected by lincRNAs and their target genes.

**Abstract:**

Visible and long-lasting estrus expression of gilts and sows effectively sends a mating signal. To reveal the roles of Long Intergenic Non-coding RNAs (lincRNAs) in estrus expression, RNA-seq was used to investigate the lincRNAs expression of follicular tissues from Large White gilts at diestrus (LD) and estrus (LE), and Chinese Mi gilts at diestrus (MD) and estrus (ME). Seventy-three differentially expressed lincRNAs (DELs) were found in all comparisons (LE vs. ME, LD vs. LE, and MD vs. ME comparisons). Eleven lincRNAs were differentially expressed in both LD vs. LE and MD vs. ME comparisons. Fifteen DELs were mapped onto the pig corpus luteum number Quantitative Trait Loci (QTL) fragments. A protein–protein interaction (PPI) network that involved estrus expression using 20 DEGs was then constructed. Interestingly, three predicted target DEGs (PTGs) (*CYP19A1* of MSTRG.10910, *CDK1* of MSTRG.10910 and MSTRG.23984, *SCARB1* of MSTRG.1559) were observed in the PPI network. A competitive endogenous RNA (ceRNA) network including three lincRNAs, five miRNAs, and five genes was constructed. Our study provides new insight into the lincRNAs associated with estrus expression and follicular development in gilts.

## 1. Introduction

Reproductive performances are important economic indicators in pig production industries. Estrus expression behavior is one of the major factors affecting the reproductive efficiency of gilts and sows. Clearly visible and long-lasting estrus expression behavior effectively sends a mating signal and improves conception rates, as well as reducing the duration of non-pregnancy of gilts and sows [1,2]. The lack of estrus expression behavior in gilts has been widely noted as a challenge to pork producers [3,4,5]. Previous studies have shown that the estrus expression of gilts could be regulated by serum estrogen because the intensity of estrus was positively correlated with serum estrogen concentration [6,7,8]. During the estrus cycle, biological processes such as follicular development, maturation, and atresia were generated alternately [9]. With the development of follicles, the theca interna secretes testosterone in response to luteinizing hormone (LH) stimulation and the granulosa cells secrete estradiol in response to follicle-stimulating hormone (FSH) stimulation. The estrus cycle was maintained by testosterone and estradiol together. Gilts in estrus show a series of estrus expression signs, such as reddening and swelling vulva, and standing reflex [10]. The strength of estrus expression is influenced by the environment and the genotypes of gilts [5]. 

Long intergenic non-coding RNAs (lincRNAs) are a class of non-coding RNAs more than 200 bp in length that do not overlap protein-coding genes [11,12,13]. lincRNAs also contain promoter- or enhancer-associated RNAs that are gene proximal and can be either in the sense or antisense orientation [14]. The length of their transcripts is shorter, the number of exons is less, and the conservation is lower in lincRNAs than in those of protein-coding genes [15,16]. lincRNAs constitute more than half of lncRNA transcripts in humans [17]. Regulation of lincRNAs includes cis- and trans-acting. In cis-acting, lincRNAs act on adjacent genes to regulate their expression, while in trans-acting lincRNAs act on genes within different strands of DNA to regulate their expression [18]. Previous studies have shown that lncRNAs bind to miRNAs as competitive endogenous RNAs (ceRNAs) to prevent miRNAs from binding to their target genes, which reduces the inhibition of miRNAs on target genes [19,20,21]. After post-transcription, lncRNAs regulate the level of mRNA by degrading mRNA [22]. As a class of lncRNAs, lincRNAs have the same function as lncRNA. More than 15,000 lincRNAs have been identified in humans, and more than 10,000 lincRNAs have been identified in mice, and these have important roles in biological processes, such as gene expression control, scaffold formation, and epigenetic control [14,23,24]. Moreover, previous studies have reported that some small molecules, such as antisense oligonucleotides (ASOs) [25] and siRNAs [26], can bind lncRNAs to regulate animal traits, which suggests that it is feasible to interfere gene expression with lincRNAs as a molecular therapy.

lincRNAs have been reported to regulate skeletal muscle growth, growth performance, intramuscular fat content, meat quality, and pre-implantation embryonic development in pigs [12,15,27,28,29,30]. For example, 120 differentially expressed lincRNAs (DELs) and 2638 differentially expressed protein-coding genes (DEGs) were associated with skeletal muscle development in pigs [27]; 759 lincRNAs and their predicted target genes were associated with the growth and meat quality differences between Yorkshire and Wannanhua pigs [28]; 1078 lincRNAs were identified in weaned piglets, and those DELs and quantitative trait loci (QTL) may play important roles in the growth and development of piglets [15]. However, follicular lincRNAs that are related to estrus traits have not been reported.

Compared with European pig breeds, Chinese indigenous pig breeds have greater litter size and a more intense and longer duration of estrus behavior expression [31]. Specifically, gilts of the Meishan breed, which originated in the Lake Taihu basin in Jiangsu (eastern China), express longer behavioral estrus and reach puberty at an earlier age than Large White and Landrace [32,33] gilts. Chinese indigenous Mi pigs also originated in the Lake Taihu basin [34]. Mi gilts have more intense indicators of estrus, such as vulva reddening and swelling, standing reflex, and mucous discharge from the vulva than Large White gilts [33,35]. The molecular regulatory mechanism that leads to the difference in estrus behavior between the two pig breeds remains unknown. We hypothesized that the genes involved in the regulation of ovarian steroid hormones were differentially expressed between the two pig breeds at different stages of the estrus cycle, and the expression of these genes was affected by lincRNAs, which further affects estrus behavior of gilts.

In the present study, RNA sequencing data were used to investigate the lincRNA expression profiles of follicular tissue from Large White gilts at diestrus (LD) and estrus (LE), and Mi gilts at diestrus (MD) and estrus (ME). The objective of this study is to identify the lincRNAs and genes that are related to estrus expression traits in gilts and to construct protein–protein interaction (PPI) and ceRNA networks that relate to estrus expression. It is possible to improve estrus behavior expression by molecular intervention using siRNA and to improve the conception rate and reproductive performance of sows. Our study provides new insights into the lincRNAs and molecular mechanisms associated with the estrus expression in pigs.

## 2. Materials and Methods

All experiments in this study were approved by the Experimental Animal Welfare and Ethics Committee of Nanjing Agricultural University, Nanjing, China (SYXK Su 2017-0027).

### 2.1. Sample Collection

Thirty Large White gilts and 30 Mi gilts were observed for the expression of estrus at Yong Kang Agricultural Science and Technology Co., Ltd. in Changzhou, Jiangsu Province, China. Estrus detection was carefully performed twice daily (at 7:00 and 15:00) at pro-estrus. On the first day of the third pubertal estrous, the day the gilts were at the onset of exhibiting standing reflex, three Large White and three Mi gilts were slaughtered humanely after anesthesia and recorded as LE (Large White gilts at estrus, n = 3) and ME (Mi gilts at estrus, n = 3), respectively. On the 10th day of the third estrous, three Large White and three Mi gilts were slaughtered in the same way and recorded as LD (Large White gilts at diestrus, n = 3) and MD (Mi gilts at diestrus, n = 3), respectively. Ovaries were dissected and all samples were collected with better ovulation points or preovulatory follicles on the surfaces of the ovaries at estrus or tissues in the same position of the ovaries at diestrus. All samples were frozen in liquid nitrogen and stored at −80 °C until RNA isolation [36]. 

### 2.2. RNA Isolation and Sequencing

Total RNA of the follicles was extracted using TRIzol reagent (Invitrogen, Carlsbad, CA, USA). Degradation and contamination of total RNA were monitored on the agarose gel. The RNA concentration and purity were checked using a Qubit RNA Assay Kit in Qubit 2.0 Fluorometer (Life Technologies, CA, USA) and a NanoPhotometer^®^ Spectrophotometer (IMPLEN, CA, USA). Furthermore, RNA integrity was assessed using an RNA Nano 6000 Assay Kit of the Agilent Bioanalyzer 2100 system (Agilent Technologies, CA, USA). The cDNA libraries were constructed using NEBNext Ultra™ RNA Library Prep Kit for Illumina (NEB, Ipswich, MA, USA). Briefly, the first-strand cDNA was synthesized using random hexamer primer and M-MuLV Reverse Transcriptase (RNase H). Subsequently, the second strand cDNA synthesis was performed using DNA polymerase I and RNase H after adding buffer, dNTPs, DNA polymerase I (New England Biolabs), and RNase H (Invitrogen). The NEBNext Adapter ligated the DNA fragments of adenylation of 3′ ends to prepare for hybridization. Lengths of 150–200 bp cDNA fragments for PCR amplification were selected to create cDNA libraries. The libraries were sequenced on an Illumina Hiseq 2500 platform, generating 125 bp paired-end reads. 

### 2.3. Quality Control, Mapping, and Transcriptome Assembly

A Mi gilt sample at diestrus was discarded because of degradation, and a Large White gilt sample at diestrus was discarded because of low quality of sequencing. Ten RNA-seq libraries of follicular tissue from five Large White gilts (2 LDs and 3 LEs) and five Mi gifts (2 MDs and 3 MEs) were generated. Clean reads were obtained using Trimmomatic [37] (version 0.39) by removing adapter sequences, low-quality reads with ploy-A/T/G/C, and poly-N sequence reads with a length less than 50 bp after being filtered and reads with Qphred ≤ 20 more than 50% nucleotides. In addition, Q20, Q30, and GC contents were calculated for the clean reads. The high-quality clean reads were used for downstream analysis. Clean reads were mapped to the pig reference genome (Sscrofa11.1, http://asia.ensembl.org/Sus_scrofa/Info/Index, accessed on 5 August 2021) by HISAT2 [38] (version 2.0.5) with default parameters, and all alignment results were assembled into one complete GTF file by StringTie [39] (version 1.3.3) for transcriptome assembly.

### 2.4. LincRNAs Identification and Characterization

The non-redundant transcriptome was processed to identify the lincRNAs based on the characteristics of lincRNAs and a series of strict screening conditions, as follows: (1) The transcripts with the ‘U’ category categorized (represent intergenic transcripts) were filtered by gffcompare in StringTie [39] (version 1.3.3); (2) The transcripts with exon number < 2, length < 200 bp were removed; (3) The coding potential of transcripts were analyzed by CPC2 [40], Pfam [41], NCBI NR, and the UniRef 90 database [42], eliminating transcripts of any know protein-coding and similar to a known protein domain. The transcripts that could not encode proteins were retained; (4) The transcripts expressed in at least one sample were reserved; those transcripts were the final candidate lincRNAs for further analysis; (5) All candidate lincRNAs were subjected to BLAST with a known pig lncRNAs database in ALDB [43]. Known and novel lincRNAs were obtained. 

We identified the protein-coding transcripts by annotating as “protein-coding” in NCBI annotation release 106 of the current pig genome reference sequence (Sscrofa11.1), then lincRNAs and protein-coding transcripts were compared via different characteristics: transcript length, exon number, exon length, and FPKM. 

### 2.5. Analysis of Differential Expression 

The number of reads across each sample was counted by the “featureCounts” program [44]. The ”DESeq2” package [45] was used to identify the differential expression of all transcripts from different time points and pig breed. A threshold “adjusted *p*-value < 0.05 and |log2 (fold change)| > 2″ was set to screen the differentially expressed transcripts (including lincRNAs and protein-coding genes) between LE vs. ME, LD vs. LE, and MD vs. ME comparisons [46]. The log2 (fold change) was calculated as log2 (FPKM_A/FPKM_B) (FPKM_A: FPKM of group LE or ME; FPKM_B: FPKM of group LD or MD or LE). The similarity of the samples based on the differential expression data was then analyzed. 

### 2.6. Functional Prediction of DELs by QTL Analysis

To predict the lincRNAs functions, all DELs were selected to perform the QTL analysis. The overlap regions of lincRNAs and QTLs on the genome was determined with BEDTools (version 2.15.0) [47], and the QTL database of the pig was downloaded from the Animal QTLdb (https://www.animalgenome.org/cgi-bin/QTLdb/SS/index, accessed on 27 August 2021) [48].

### 2.7. Target Genes Prediction and Function Enrichment Analysis

To further explore the lincRNAs functions, the target genes by cis- and trans-acting of the DELs were predicted. First, the protein-coding genes within 100 kb upstream and downstream of lincRNAs were identified by BEDTools (version 2.15.0) [47], and the target genes by cis-acting of the DELs were predicted [49]. Then, based on the correlated expression levels with DELs, the protein-coding genes whose Pearson correlation coefficients were more than 0.95 were selected, and the target genes by trans-acting of the DELs were predicted [50]. Gene Ontology (GO) enrichment and the Kyoto Encyclopedia of Genes and Genomes (KEGG) pathway analyses of the predicted target DEGs (PTGs) and DEGs were performed by the DAVID database (https://david.ncifcrf.gov/tools.jsp, accessed on 10 September 2021). A *p*-value less than 0.05 was considered statistically significant in terms of GO enrichment and KEGG pathways [51]. A PPI analysis of the genes involved in estrus expression was performed using the STRING database; these genes were enriched into KEGG pathways related to steroid biosynthesis and follicular development. For further visualization, the PPI and the co-expression network between DELs and their PTGs were analyzed using Cytoscape (https://cytoscape.org, accessed on 11 October 2021).

### 2.8. Construction of ceRNA Network Related to Follicular Development and Estrus Expression

lncRNAs regulate the expression of mRNAs by competitively binding to the same miRNAs [52]. To determine the interactions between lincRNAs and mRNAs, miRanda software was used to analyze the interaction between lincRNAs and miRNAs and between miRNAs and mRNAs. The porcine miRNA sequences were downloaded from the miRBase database (v.22.1, http://www.mirbase.org, accessed on 24 September 2021). DELs that were differentially expressed in LD vs. LE and MD vs. ME comparisons, and their PTGs that were involved in follicular development and estrus expression, were selected, and only remained in the alignments with a comprehensive score greater than 150, with no mismatch in the positions. The potential regulatory network was visualized using Cytoscape (https://cytoscape.org, accessed on 11 October 2021).

### 2.9. RT-qPCR Verification

To verify the results of RNA sequencing, six samples were selected from the LD, LE, MD, and ME groups to perform RT-qPCR. Total cDNA was synthesized using a reverse transcriptase Kit (TaKaRa, Dalian, China). RT-qPCR was performed on a QuanuStudio 5 using SYBR Green Master Mix (Vazyme Biotech, Nanjing, China). PCR reactions were performed in triplicate and primers were designed by the Primer Premier 5 program (Appendix A). Four DELs and their PTGs were randomly selected and determined by RT-qPCR. Real-time PCR reactions were performed on an Applied Biosystems Step One Plus system using the following program: 10 min at 95 °C, followed by 40 cycles of 15 s at 95 °C and 60 s at 60 °C, then a 60–95 °C melting curve detection. The relative expression of lincRNAs and protein-coding genes were calculated by using the 2^−∆∆Ct^ method and normalized by *GAPDH*. The data of expression were analyzed by GraphPad Prism (version 8.0, San Diego, CA, USA). A Pearson correlation was performed to determine the correlation between lincRNAs and PTGs. The tests were considered significant at *p*-value < 0.05. Results were reported as means ± standard error of the mean (SEM).

## 3. Results

### 3.1. Identification and Characterization of lincRNAs

A total of 513,109,520 clean reads from 10 follicular tissue samples (2 LDs, 3 LEs, 2 MDs, and 3 MEs) were retained after removing the adaptor and low-quality reads (Appendix A). We constructed a pipeline to identify the lincRNAs (Figure 1A), and 337 putative lincRNAs were identified (Appendix A), including 138 known lincRNAs and 199 novel lincRNAs (Figure 1B). Those putative lincRNAs were mainly distributed on chromosomes 1, 6, and 9 (Figure 1C). Overall, a total of 17,173 protein-coding genes were identified.

The expression levels of known and novel lincRNAs were less than protein-coding genes (Figure 1D). The average length of transcripts in the novel lincRNAs (738.87 ± 50.67) and known lincRNAs (1232.40 ± 107.99) was less than that in the protein-coding genes (3662.72 ± 10.70) (Figure 1E); while the length of exons in the novel lincRNAs (317.95 ± 20.98) and known lincRNAs (448.92 ± 39.09) was greater than that in the protein-coding genes (278.60 ± 0.80) (Figure 1F). Furthermore, the number of exons in the novel lincRNAs (2.34 ± 0.04) and known lincRNAs (2.74 ± 0.11) were less than that in the protein-coding genes (13.10 ± 0.05) (Figure 1G). 

### 3.2. Analysis of Differentially Expressed LincRNAs (DELs) 

Based on the differential expression levels, we obtained 73 DELs by the screening criteria: |log2 (fold change)| > 2 and p-adjusted < 0.05 (Figure 2D). Specifically, there were 30, 37, and 29 DELs detected in LE vs. ME, LD vs. LE, and MD vs. ME comparisons, respectively (Figure 2). Simultaneously, we detected 620, 1539, and 1597 DEGs in LE vs. ME, LD vs. LE, and MD vs. ME comparisons, respectively (Appendix A). The hierarchical clustering analyses of DELs showed clear breed-specificity in LE vs. ME comparison (Figure 2E), time-specificity in LD vs. LE (Figure 2F), and MD vs. ME comparisons (Figure 2G). The hierarchical clustering analyses of DEGs showed the same results as those of DELs (Appendix A).

### 3.3. Target Gene Prediction and Function Analysis of lincRNAs

To obtain the potential target genes near DELs, the genes located upstream and downstream of DELs within 100 kb were identified. A total of 131 PTGs were expressed in at least one sample (Appendix A). The results of GO enrichment analysis of these 131 PTGs of the DELs showed that the greatest enrichment was the biological process term (Figure 3A). Furthermore, 418 PTGs were obtained by trans-acting the DELs. The biological process term was the greatest enrichment term. Notably, some terms were related to a steroid hormone, such as “response to steroid hormone” in biological process, and “steroid-binding” in molecular function (Figure 3B).

Then, lincRNAs differentially expressed in all comparisons were analyzed. lincRNAs MSTRG.15572 and MSTRG.24167 were observed (Figure 3C). Furthermore, 3 PTGs regulated by cis-acting and 14 PTGs regulated by trans-acting of MSTRG.24167 were predicted, and 5 PTGs regulated by trans-acting of MSTRG.15572 were predicted. Simultaneously, a co-expression network including those 2 key lincRNAs and their PTGs was constructed (Figure 3D). 

### 3.4. Analysis of DELs by QTLs

DELs were mapped onto the porcine QTL database to explore their function. A total of 2041 QTLs were observed, including 5 trait categories, 27 trait types, and 345 traits. The 5 trait categories consist of 115 exterior, 264 health, 1355 meat, 194 production, and 112 reproduction QTLs (Figure 4D). These QTLs were mainly distributed on chromosomes 1, 6, and 7 (Figure 4A). Furthermore, the function of 112 reproduction QTLs were analyzed. These reproduction QTLs were mainly distributed on chromosomes 1, 6, and 10 (Figure 4B). Interestingly, 16 QTLs of corpus luteum number were observed (Figure 4C). Simultaneously, 15 DELs were mapped onto the pig corpus luteum number QTL fragments (Appendix A). Then, 22 PTGs regulated by cis-acting and 55 PTGs regulated by trans-acting in 15 DELs were predicted in a co-expression network (Figure 4E). Surprisingly, MSTRG.24167 had been observed again in this analysis. MSTRG.21086 and its trans-acting PTG *KIF16B* were differentially expressed in both LD vs. LE and MD vs. ME comparisons.

### 3.5. Analysis of DELs by Differentially Expressed Protein-Coding Genes (DEGs)

We detected 2553 DEGs between all comparisons (Appendix A). GO enrichment and KEGG pathway analyses of DEGs were performed. In the GO enrichment analysis of 620 DEGs in LE vs. ME comparison, 69 terms were significant (*p* < 0.05); 153 terms were obtained in the GO enrichment analysis of 1539 DEGs in LD vs. LE comparison; 181 terms were obtained in the GO enrichment analysis of 1597 DEGs in MD vs. ME comparison (Appendix A).

Simultaneously, seven KEGG pathways were obtained in LE vs. ME comparison (Appendix A). Thirty-three pathways were obtained in LD vs. LE comparison (Figure 5A). Thirty-seven pathways were obtained in MD vs. ME comparison (Figure 5B). Interestingly, 31 DEGs were enriched in pathways involved in estrus expression in LD vs. LE comparison, including the steroid hormone biosynthesis, steroid biosynthesis, progesterone-mediated oocyte maturation, and ovarian steroidogenesis pathways (Figure 5C). Thirty-five DEGs were enriched in pathways related to estrus expression in MD vs. ME comparison, including ovarian steroidogenesis, steroid hormone biosynthesis, steroid biosynthesis, ovarian steroidogenesis, and oocyte meiosis pathways (Figure 5D). Moreover, these DEGs involved in estrus expression were the PTGs regulated by trans-acting DELs. The DELs predicted in LD vs. LE and MD vs. ME comparisons are shown in Table 1. A total of 20 DEGs were enriched in pathways involved in estrus expression in LD vs. LE and MD vs. ME comparisons (Figure 5E). The PPI network analysis showed that *SQLE* and *CYP19A1* genes were the key nodes (Figure 5F). Surprisingly, some PTGs of DELs by trans-acting, such as *CYP19A1*, *CDK1*, and *SCARB1*, were observed in this PPI network (Figure 5F, Table 1). Interestingly, MSTRG.1559, MSTRG.10910, and MSTRG.23984 were predicted in LD vs. LE and MD vs. ME comparisons. MSTRG.6832 was the specific lincRNA in the LD vs. LE comparison, and MSTRG.24167 was the specific lincRNA in the MD vs. ME comparison. 

### 3.6. Construction of the lincRNA-miRNA-mRNA Network

Differentially expressed DELs in LD vs. LE and MD vs. ME comparisons and their PTGs were selected to analyze their interactions using the miRanda database. As a result, a ceRNA network involved in estrus expression was constructed (3 DELs, 5miRNAs, and 5 DEGs) (Figure 6A). In this ceRNA network, MSTRG.23984, MSTRG.21086, and their PTGs (*CDK1*, *CCNE2*, *SGO1*, and *KIF16B*) were differentially expressed in LD vs. LE and MD vs. ME comparisons. MSTRG.6832 and its PTG (*CCNA1*) were differentially expressed in the LD vs. LE comparison (Table 1). These results suggest that estrus expression may be regulated by lincRNAs, which play important roles in estrus expression and the follicular development of pigs (Figure 6B).

To verify the results of RNA-Seq, 4 DELs and their PTGs were randomly selected and determined by RT-qPCR. In the data of RNA-Seq, Pearson correlation coefficients of four pairs of genes were greater than 0.90 or less than −0.90, and the *p*-value was less than 0.01 in six samples. The results of RT-qPCR showed that Pearson correlation coefficients of 4 pairs of genes were greater than 0.80 or less than −0.80, and the *p*-value was less than 0.05 in six samples (Figure 6C–F). Therefore, the results of RT-qPCR are consistent with the results of RNA-Seq. 

## 4. Discussion

Female pigs are animals with perennial estrus, and usually show a series of behavioral characteristics at estrus, such as reddening and swelling of the vulva, mucous discharge from the vulva, and standing reflex. Lack of estrus behavioral expression decreases the accuracy of heat detection, which leads to decrease in conception rates. Previous studies have performed RNA-seq analysis to identify the genes and alternative splicing that affect porcine follicular development [50,53,54]. Another previous study has investigated lncRNAs in medium-sized ovarian follicles that contributed to developmental differences between Meishan and Duroc sows [55]. Except for our previous study [36], few studies have investigated the differentially expressed genes between different stages of the estrus cycle and gilts of different pig breeds [56]. In particular, the effects of lincRNAs on estrus expression in pigs have not been reported. In this study, the follicular tissues of Large White and Mi gilts at estrus and diestrus were collected to perform RNA-sequencing, which explored the associations of lincRNAs in the development of follicles and estrus expression. 

In mammals, most lncRNAs are lincRNAs [13,14], which have been identified in humans and mice [57]. Even though pigs and humans are highly homologous, the expression of lincRNAs is species and spatio-temporal specific. The average length of the transcript is shorter, the average length of the exon is longer, the number of exons is less, and the expression level is less in the lincRNA genes than those in the protein-coding genes in pigs [12,15,27]. In our present study, a total of 337 lincRNAs were observed, and the characteristics of these lincRNAs are consistent with previous studies [58,59]. Interestingly, the known lincRNA genes have fewer exons and longer transcripts than those of the novel lincRNA genes. This may be related to the small sample size of the present study, and also indicates that porcine lincRNAs deserve further study.

In the present study, 12 lincRNAs were specifically expressed in Mi gilts, 18 lincRNAs were specifically expressed in Large White gilts, and 19 lincRNAs were specifically expressed in Large White and Mi gilts at estrus. These results indicate that the expression of lincRNAs was highly specific in each pig breed and stage of the estrus cycle; lincRNAs regulate gene expression levels by cis- and trans-acting [27,60]. GO enrichment and KEGG pathway analyses identified 131 PTGs regulated by cis-acting and 418 PTGs regulated by trans-acting in biological processes, for example, regulation of cell shape. Consistently, a previous study found that lincRNAs are involved in biological processes [22]. Steroids may be involved in the control of oocyte maturation [61]. In fact, complete estrus expression is regulated by the action of both the progesterone and estrogen systems, and these ovarian steroid hormones have an impact on the intensity and duration of estrus expression [62]. A previous study has shown that progesterone decreases and estradiol increases before ovulation [63]. E2/P4 begins to decrease during ovulation, along with the decline in the intensity and duration of estrus [64], suggesting that both progesterone and estradiol play important regulatory roles in estrus ovulation. Notably, PTGs regulated by trans-acting were enriched in two terms related to the regulatory function of steroid hormones: response to steroid hormone and steroid-binding. Moreover, MSTRG.24167 and MSTRG.15572 were differentially expressed in LE vs. ME, LD vs. LE, and MD vs. ME comparisons, which may play an important role in the estrus expression of pigs. Three PTGs (*PLET1*, *PTS*, and *BOC2*) by cis-acting of MSTRG.24167 were predicted; the *PLET1* gene has been reported as being expressed in placentas of pigs and mice [65,66], and the *BCO2* gene has been reported as a possible candidate gene for identifying composite reproductive traits of Lori-Bakhtiari sheep [67]. In 22 PTGs predicted by MSTRG.24167 and MSTRG.15572, in the present study several genes, such as *CAMK2A* [68], *LRP11* [69], and *LPAR3* [70], are involved in ovarian function. Fifteen DELs were mapped onto the pig corpus luteum number QTL fragments. Some PTGs of these 15 DELs, such as *PLET1* [65], *CCDC141* [71], and *KIF16B* [72], are associated with reproduction traits. These results imply that lincRNAs regulate their PTGs by cis- and trans-acting, which is associated with follicular development and estrus behavioral expression. 

Previous studies showed that the PI3K-Akt signaling pathway activated the Wnt signaling pathway, and the Wnt signaling pathway was related to the apoptosis of ovarian granulosa cells in pigs [50,73]. The DEGs in the LD vs. LE and MD vs. ME comparisons were significantly enriched in the PI3K-Akt signaling pathway and metabolic pathways in the present study. Interestingly, four and five pathways related to ovarian function and estrus expression were significantly enriched in the LD vs. LE and MD vs. ME comparisons, respectively. Then some PTGs of DELs were observed in a PPI network, which is involved in estrus expression. Because the estrogen synthesis has reached its peak due to the arrival of the luteinizing hormone (LH) peak on the day before estrus, the expression level of genes involved in estrogen synthesis, such as the *CYP19A1* gene, begins to decline [74]. Then the expression level of the *PTGS* gene related to ovulation begins to up-regulate to promote vascular dilation [64]. This is also consistent with the sequencing results in the present study. It indicates that the samples we used for investigation are closely related to follicular development and estrus expression; these identified DEGs and DELs play an important role in regulating estrus expression. In addition, whether or not the DEGs that are not related to ovarian steroid biosynthesis in the LD vs. LE and MD vs. ME comparisons affect estrus behavior in gilts through other means needs further study. 

Interestingly, we found that some DELs are related to cholesterol anabolism in the LE vs. ME comparison. For example, *LDLR*, *HMGCR*, and *STARD4* have been shown to play important roles in de novo cholesterol uptake [75], rate-limiting [76], and transport [77]. Cholesterol is the substrate for all steroid hormones, suggesting that cholesterol is essential for estrus behavior in gilts [78,79]. In our sequencing data, some PTGs of DELs were enriched on the steroid biosynthesis pathway. These genes are associated with steroid uptake, mitosis, and estradiol biosynthesis. Therefore, the specific lincRNA MSTRG.6832 and its target gene *CCNA1* were differentially expressed in the LD vs. LE comparison. *CCNA1*, a meiosis-specific cyclin, may have functions in meiosis distinct from their mitotic functions [80,81]. The specific lincRNA MSTRG.24167 and its target gene *CAMK2A* were found in the MD vs. ME comparison. The expression level of MSTRG.24167 was greater at the estrus than at the diestrus of gilts. *CAMK2A* was significantly enriched in the oocyte meiosis pathway. It has been reported that phosphorylates EMI2 and WEE1B inactivated the M-phase promoting factor protein kinase activity (MPF), and this ultimately triggers meiotic resumption [68,82]. Moreover, MSTRG.1559 and MSTRG.10910 were differentially expressed in LD vs. LE and MD vs. ME comparisons, MSTRG.23984 and MSTRG.21086, and their PTGs (*CDK1*, *CCNE2*, *SGO1*, *KIF16B*) were differentially expressed in the LD vs. LE and MD vs. ME comparisons. Then, a ceRNA network showed that those DELs interact with DEGs through miRNAs. It is important to note that the gilts in this study were selected from Large white and Mi gilts with clearer estrus behavior in the groups. This may have some differences and limitations because of small sample size, but the DELs and DEGs identified by comparing the differences between two pig breeds and two stages of the estrus cycle also provide important information for us to understand the genetic mechanism of estrus. We hypothesized that those DELs and DEGs are important to follicular development and estrus behavioral expression in gilts. Therefore, the specific regulatory mechanism of DELs is worthy of further study.

## 5. Conclusions

In conclusion, we investigated the lincRNAs and protein-coding genes of Large White and Mi gilts at estrus and diestrus, identified the differential expression of lincRNAs, and constructed a ceRNA network including three lincRNAs, five miRNAs, and five protein-coding genes that related to follicular development and estrus expression. Functional analysis showed that target genes of differential expression lincRNAs were involved in signaling pathways and terms related to ovarian function. Importantly, some lincRNAs are differentially expressed in these comparisons, and their target genes are involved in the regulation of follicular development and estrus expression, such as MSTRG.1559, MSTRG.6832, MSTRG.10910, MSTRG.21086, MSTRG.23984, and MSTRG.24167. These results suggest that lincRNAs regulate their target genes, which are associated with estrus behavior expression and the regulatory mechanism needs to be further studied. Our study identified the lincRNAs associated with estrus expression and provides new insight into the molecular mechanisms of estrus expression and follicular development in gilts.

## Figures and Tables

**Figure 1 biology-11-00716-f001:**
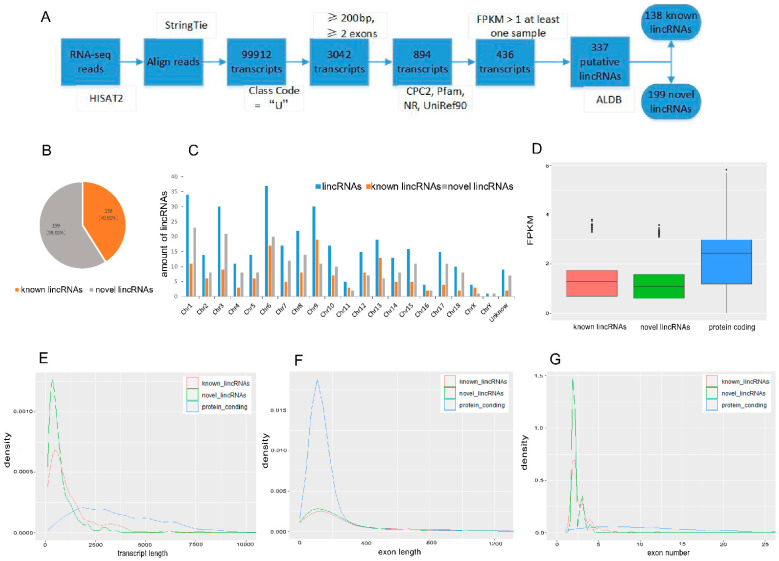
Identification and characterization of lincRNAs and protein-coding genes. (**A**) The pipeline to identify the lincRNAs. (**B**) The number of novel and known lincRNAs. (**C**) Distribution of lincRNAs on chromosomes. (**D**) FPKM of the known lincRNAs, novel lincRNAs, and protein-coding genes. (**E**–**G**) Distribution of transcript length, exon length, and exon number of the known lincRNAs, novel lincRNAs, and protein-coding genes. CPC: Coding Potential Calculator; NR: non-redundant database; FPKM: fragments per kilobase of transcript per million mapped reads.

**Figure 2 biology-11-00716-f002:**
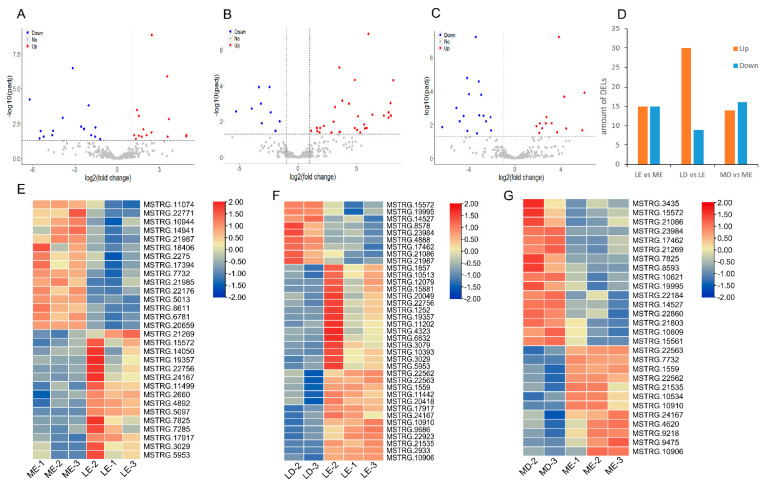
Analysis of differentially expressed lincRNAs. (**A**–**C**) Screening of the DELs between LE vs. ME, LD vs. LE, and MD vs. ME comparisons. (**D**) The number of DELs. (**E**–**G**) Hierarchical clustering heatmap of DELs. The color scale represents gene expression levels. Each row represents one gene, and each column represents one sample.

**Figure 3 biology-11-00716-f003:**
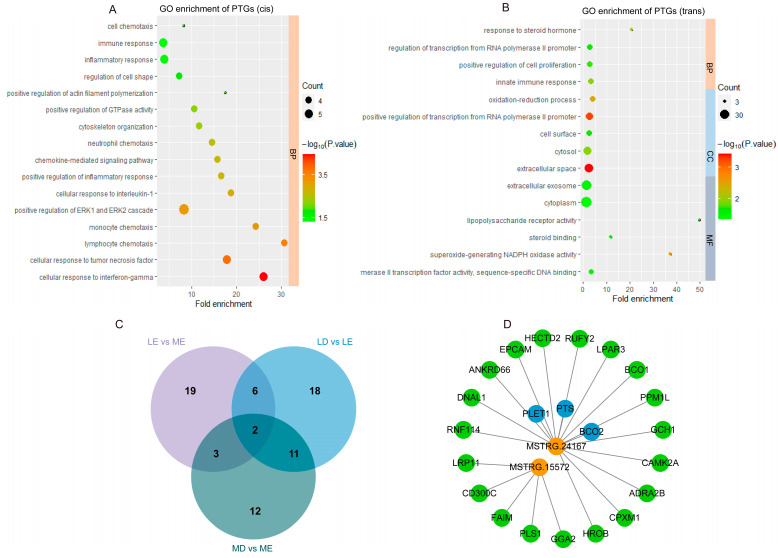
Function analysis and identification of key lincRNAs. (**A**) GO enrichment analysis of PTGs by cis-acting of DELs. (**B**) GO enrichment analysis of PTGs by trans-acting of DELs. (**C**) DELs distribution in the LE vs. ME, LD vs. LE, and MD vs. ME comparisons. (**D**) Co-expression network of the 2 key lincRNAs and their PTGs; the blue represents PTGs by cis-acting, and the green represents PTGs by trans-acting.

**Figure 4 biology-11-00716-f004:**
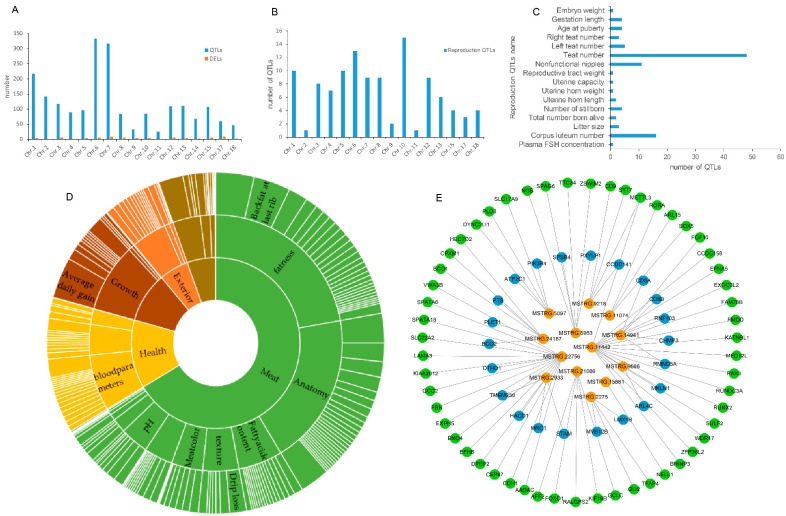
Function analysis of lincRNAs by QTLs. (**A**) Distribution of the DELs and QTLs on chromosomes. (**B**) Distribution of the reproduction QTLs on chromosomes. (**C**) The number of the reproduction QTLs. (**D**) Overview of the QTLs that were mapped onto the DELs. (**E**) Co-expression network of the DELs and their PTGs; the blue represents PTGs by cis-acting, and the green represents PTGs by trans-acting.

**Figure 5 biology-11-00716-f005:**
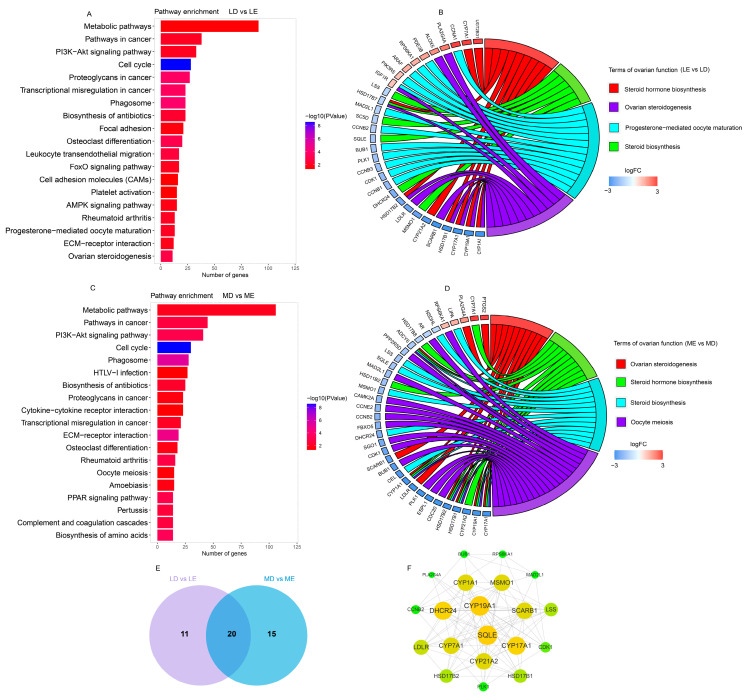
Function analysis of differentially expressed protein-coding genes. (**A**) KEGG pathway analysis of DEGs between LD vs. LE comparison. (**B**) KEGG pathway analysis of DEGs between MD vs. ME comparison. (**C**) Overview of the DEGs and pathways related to follicular development and estrus expression between LD vs. LE comparison. (**D**) Overview of the DEGs and pathways related to follicular development and estrus expression between MD vs. ME comparison. (**E**) Overlap analysis of the genes that related to follicular development and estrus expression between LD vs. LE and MD vs. ME comparisons. (**F**) PPI network of the genes that related to follicular development and estrus expression between LD vs. LE and MD vs. ME comparisons. The size of the circle represents the degree of interaction between the genes.

**Figure 6 biology-11-00716-f006:**
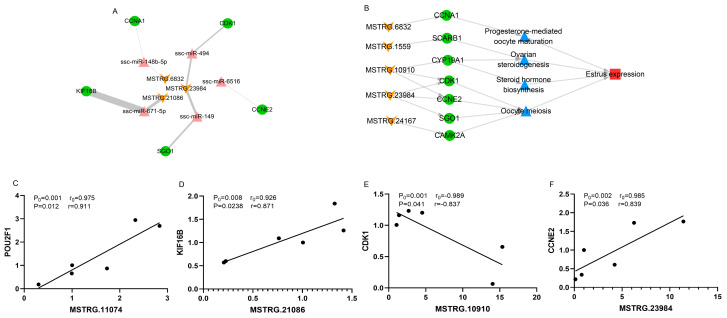
ceRNA network and the mechanism of DELs regulating estrus expression. (**A**) ceRNA network of DELs and DEGs. The yellow quadrilateral represents DELs, the red triangle represents miRNA, the blue circle represents DEGs, and the thickness of the edge represents energies. (**B**) The mechanism of DELs regulating estrus expression. The yellow quadrilateral represents DELs, the blue circle represents DEGs, and the blue triangle represents pathways. (**C**) Linear regression of MSTRG.11074 and *POU2F1* in six samples. (**D**) Linear regression of MSTRG.21086 and *KIF16B* in six samples. (**E**) Linear regression of MSTRG.10910 and *CDK1* in six samples. (**F**) Linear regression of MSTRG.23984 and *CCNE2* in six samples. r_0_ and P_0_ represent the Pearson correlation coefficient and *p*-value by RNA-seq, and r and P represent verification by RT-qPCR.

**Table 1 biology-11-00716-t001:** The correlation between DELs and the PTGs that are involved in estrus expression.

Comparison	DELs	PTGs	Desription	KEGG Pathway	r	*p*
MD vs. ME	MSTRG.10910	*CCNE2*	cyclin E2	Oocyte meiosis	−0.96	<0.01
	MSTRG.23984	*CCNE2*			0.97	<0.01
	MSTRG.1559	*SCARB1*	scavenger receptor class B member 1	Ovarian steroidogenesis	−0.95	<0.01
	MSTRG.10910	*CDK1*	cyclin dependent kinase 1	Oocyte meiosis	−0.96	<0.01
	MSTRG.23984	*CDK1*			0.96	<0.01
	MSTRG.23984	*SGO1*	shugoshin 1	Oocyte meiosis	0.96	<0.01
	MSTRG.24167	*CAMK2A*	calcium/calmodulin dependent protein kinase II alpha	Oocyte meiosis	−0.96	<0.01
	MSTRG.10910	*CYP19A1*	cytochrome P450 19A1	Steroid hormone biosynthesis, Ovarian steroidogenesis	−0.96	<0.01
LD vs. LE	MSTRG.6832	*CCNA1*	cyclin A1	Progesterone-mediated oocyte maturation	0.95	<0.01
	MSTRG.1559	*SCARB1*	scavenger receptor class B member 1	Ovarian steroidogenesis	−0.95	<0.01
	MSTRG.10910	*CDK1*	cyclin dependent kinase 1	Progesterone-mediated oocyte maturation	−0.96	<0.01
	MSTRG.23984	*CDK1*			0.96	<0.01
	MSTRG.23984	*SGO1*	shugoshin 1	Oocyte meiosis	0.96	<0.01
	MSTRG.23984	*CCNE2*	cyclin E2	Oocyte meiosis	0.97	<0.01
	MSTRG.10910	*CYP19A1*	cytochrome P450 19A1	Steroid hormone biosynthesis, Ovarian steroidogenesis	−0.96	<0.01

r indicates Pearson correlation coefficients between DELs and their PTGs in the data of RNA-Seq.

## Data Availability

The datasets generated and/or analyzed during the current study are available from the corresponding author upon reasonable request.

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
