# Peer review of "Comprehensive Transcriptome Analysis of Follicles from Two Stages of the Estrus Cycle of Two Breeds Reveals the Roles of Long Intergenic Non-Coding RNAs in Gilts"

_biology, 2022, doi:10.3390/biology11050716_

Round 1

Reviewer 1 Report

The authors investigated the lincRNAs profile of two breeds of pigs in the ovarian tissue during diestrus and estrus phases.

The work depends on bioinformatic analysis of the RNA seq data and the authors verified some of DEGs by RT-qPCR.

There are some critical points that should be carefully revised and clarified:

1- The sample collection:

Samples are not well described. What is meant by follicle? do you mean theca cells-enclosed follicles or with follicular fluid? What is the diameter of the follicles? how many follicles did you dissect per ovary? How did you isolate follicles from the surrounded ovarian tissue? I cannot understand this main point.

2- Methods:

-How do you use GAPDH as a reference gene for LincRNAs? Why didn't you use another stable lincRNA as a housekeeping gene?

-Correct the name  (2^-ddct).

-Write details about the statistical analysis of fold-change and the correlation.

3- Results:

-You measured three and sometimes two samples, but the values are without SEM or SD.

-How about the relative gene expression of POU2F1, KIF16B, CDK1, and CCNE2 between estrus and diestrus in the two breeds?

4- Discussion, conclusion & abstract:

-You didn't measure expression corpus luteum. As you stated you measured the follicles of diestrus?!

-You have not found any link with animal behavior. Focus on your findings only.

5- References:

-Important related references are not cited:

Li et al. 2021: https://doi.org/10.1038/s41598-021-01817-y

Yang et al. 2018: https://doi.org/10.1155/2018/9150723

6- Language: 

-The manuscript contains several grammatical errors and requires extensive editing for clarity.

Reviewer 2 Report

Liu et al.

Comprehensive transcriptome analysis of follicles from two stages of the estrus cycle of two breeds reveals the roles of Long Intergenic Non-coding RNAs in gilts

Major (General or Overall) Comments to the Authors

The authors set out to identify the lincRNA expression profiles of follicular tissue from Large White and MI gilts at diestrus and estrus. Although the objective of the study is clearly stated, the authors did not present a clear hypothesis. In addition, the relevancy of investigating lincRNA expression in gilts is not well justified. The authors describe the importance of estrus expression on pig production, but how does learning from lincRNA expression contributes to improving pig production? Generating new knowledge about a specific topic does not always translate into applicable methods to improve animal production. Please consider expanding on this issue and provide specific examples on how the knowledge generated from your study can be applied into enhancing pork production.

Specific comments by section.

Introduction

As mentioned above, please clearly state your hypothesis and expand on how this knowledge can be applied into improving the pork industry.

Materials and Methods

Although the Materials and Methods section is well written and includes a detailed description of the methods used, the experimental design and sample size used in this study provide some major limitations that decrease the value of the knowledge generated from this study. First, since the animals were slaughtered (i.e., samples of both estrus and diestrus stages were not obtained from the same animal), the effect of the stage can be confounded by inherent differences between the individual animals used in the present study. On top of that, the sample size (n = 3) for each group is very limited, which increases the chances of having a confounding effect. Moreover, it is important to consider that this is an observational study (and not a randomized controlled study), which by itself already has some limitations that affect the conclusions that can be made from the results obtained. Unfortunately, the authors did not address any of these limitations in their Discussion section. This reviewer considers that although the manuscript is well written and provides some valuable information, the authors need to address and discuss these important limitations.

Results

The authors did a great job in describing the results found; the section is well written and includes very nice and descriptive Figures. Nevertheless, this section is too long, which makes it very hard to follow. Please consider removing some of the supplemental information provided and be more concise on the results presented overall (i.e., only include the most relevant results/information). In addition, font size of Figure 4 is very small, so please consider increasing its size.

Discussion/Conclusion

As mentioned above, please discuss the limitations of your study. In addition, please consider revising some of the wording used in the discussion; since this was an observational study, the authors have explored/identified the associations and NOT the effect/function of lincRNAs have on estrus expression (e.g., Ln 377-380). Please also revise spelling/grammar here and elsewhere in manuscript (e.g., Ln 428; beginning of sentence).

Round 2

Reviewer 1 Report

The manuscript has been corrected.

Reviewer 2 Report

The authors have followed all of my suggestions and have significantly improved their manuscript. Great job!